# COVID-19 vaccine acceptance among health care workers in Africa: A systematic review and meta-analysis

**Martin Ackah**[1,2]*, **Louise Ameyaw**[2,3], **Mohammed Gazali Salifu**[2,4], **Delali Pearl Afi Asubonteng**[5], **Cynthia Osei Yeboah**[1], **Eugene Narkotey Annor**[6], **Eunice Abena Kwartemaa Ankapong**[7], **Hosea Boakye**[8]

1 Department of Physiotherapy, Korle Bu Teaching Hospital, Accra, Ghana, 2 Department of Epidemiology, School of Public Health, College of Health Sciences, University of Ghana, Accra, Ghana, 3 Department of Medicine, Achimota Hospital, Accra, Ghana, 4 Policy Planning Budgeting Monitoring and Evaluation Directorate, Ministry of Health, Accra, Ghana, 5 Department of Physiotherapy, GA East Municipal Hospital, Accra, Ghana, 6 School of Public Health, East Tennessee State University, Johnson City, Tennessee, United States of America, 7 Department of Occupational Therapy, School of Biomedical and Allied Health Science, University of Ghana, Accra, Ghana, 8 Department of Physiotherapy, LEKMA Hospital, Accra, Ghana

* martinackah10@gmail.com

**Data Availability Statement:** All relevant data are within the paper and its Supporting information files.

## Abstract

### Introduction

Coronavirus Disease (COVID-19) vaccine acceptance, and hesitancy amongst Health Care Workers (HCWs) on the African continent have been examined through observational studies. However, there are currently no comprehensive reviews among these cadre of population in Africa. Hence, we aimed to review the acceptance rate and possible reasons for COVID-19 vaccine non-acceptance/hesitancy amongst HCWs in Africa.

### Methods

We searched Medline/PubMed, Google Scholar, and Africa Journal Online from January, 2020 to September, 2021. The Newcastle-Ottawa Quality Assessment tool adapted for cross-sectional studies was used to assess the quality of the retrieved studies. DerSimonian and Laird random-effects model was used to pool the COVID-19 vaccine acceptance rate. Sub-group and sensitivity analyses were performed. Reasons for COVID-19 vaccine hesitancy were also systematically analyzed.

### Results

Twenty-one (21) studies were found to be eligible for review out of the 513 initial records. The estimated pooled COVID-19 vaccine acceptance rate was 46% [95% CI: 37%-54%]. The pooled estimated COVID-19 vaccine acceptance rate was 37% [95% CI: 27%-47%] in North Africa, 28% [95% CI: 20%-36%] in Central Africa, 48% [CI: 38%-58%] in West Africa, 49% [95% CI: 30%-69%] in East Africa, and 90% [CI: 85%-96%] in Southern Africa. The estimated pooled vaccine acceptance was 48% [95% CI:38%-57%] for healthcare workers, and 34% [95% CI:29%-39%] for the healthcare students. Major drivers and reasons were

**Funding:** The author(s) received no specific funding for this work.

**Competing interests:** The authors have declared that no competing interests exist.

the side effects of the vaccine, vaccine's safety, efficacy and effectiveness, short duration of the clinical trials, COVID-19 infections, limited information, and social trust.

## Conclusion

The data revealed generally low acceptance of the vaccine amongst HCWs across Africa. The side effects of the vaccine, vaccine's safety, efficacy and effectiveness, short duration of the clinical trials, COVID-19 infections, limited information, and social trust were the major reasons for COVID-19 hesitancy in Africa. The misconceptions and barriers to COVID-19 vaccine acceptance amongst HCWs must be addressed as soon as possible in the continent to boost COVID-19 vaccination rates in Africa.

## Introduction

The current Coronavirus Disease (COVID-19) pandemic is a global public health emergency that offers significant challenges to health-care systems [1, 2]. "Coronaviruses are large, enveloped, positive-strand RNA viruses that can be categorized into genera; alpha, beta, delta and gamma, of which alpha and beta are known to infect humans" [3]. Human Corona Viruses (HCoVs) i.e. HCoV 229E, NL63, OC43 and HKU1 are endemic globally and account for 10%-30% of upper respiratory tract infections in adults humans [3].

The current basic reproductive number (R0) of Severe Acute Respiratory Syndrome Coronavirus-2 (SARS COV-2) is estimated to be three and as a result the threshold of herd immunity for COVID-19 is roughly around 67 percent [4, 5]. This purport that after the population's acquired immunity reaches 67 percent and above, COVID-19 infection rates will start to decline [6].

Individual and community initiatives such as enhanced hand cleanliness, physical distancing, and the personal protective equipment are currently being used to reduce disease transmission. However, with the world facing an economic downturn and an uncertain future, a COVID-19 vaccine is perhaps the best option for halting the epidemic [7, 8].

The SARS Cov-2 Development and Access Strategy established by Africa Center for Disease Control in 2020 aim to vaccinate at least 60% of African Population by 2022 to develop herd immunity [9]. Africa has received approximately 143 million doses in total as of September, 2021, but only 39 million people, or around 3% of the continent's population, had been adequately vaccinated. In the United States, 52 percent of people are fully vaccinated, whereas in the European Union, 57 percent are [10]. The willingness of Health Care Workers (HCWs) to be vaccinated against COVID-19 acts as a valuable role model for the general public [11].

As the vaccine becomes more widely available in Africa, Sevidzem et al identified and evaluated some probable link to vaccination acceptability in Africa. The factors included vaccination deployment plans, religious practices, vaccine hesitation, proliferation of misinformation, HCW attitudes towards the vaccine, social effects, and supportive environment [12]. Vaccine aversion among the general public has a direct association to vaccine hesitancy among HCWs [13]. Thus, HCWs role in vaccine acceptability cannot be underestimated as a result of their modeling behavior [13].

A rapid systematic review of global vaccine acceptance among HCWs ranged from approximately 28% to 73% [6]. Similarly, a comprehensive review and meta-analysis of cross-sectional studies of health workers' intentions to vaccinate against COVID-19 indicated a moderate acceptance rate [i.e., 51 percent]. The authors did admit, however, that the population studied

were largely from economically developed countries, which limited the study's generalizability [14]. Clearly, this cannot be extended to represent HCW intentions to vaccinate against COVID-19 in Africa.

COVID-19 vaccine acceptance, and hesitancy amongst HCWs on the African continent have been examined through observational studies [7, 15]. However, there are currently no comprehensive reviews among these cadre of population in Africa. Hence, we aimed to systematically review the acceptance rate and possible reasons for COVID-19 Vaccine non-acceptance/hesitancy amongst HCWs in Africa. The outcome would enable stakeholders [i.e., policy makers, researchers and government] package effective health promotion measures to boost COVID-19 vaccine uptake in Africa.

## Specific objectives

1. To determine the level of COVID-19 vaccine acceptance among HCWs in Africa.

2. To assess the drivers of COVID-19 vaccine non-acceptance/hesitancy among HCWs in Africa.

## Methods

### Protocol registration and best practice

The Center for Reviews and Dissemination standards were followed in preparing this systematic review and meta-analysis [16]. Also, the current review was conducted and reported according to the guidelines of the Preferred Reporting Items for Systematic Review and Meta-Analysis (PRISMA) [17] [see S1 Table]. The protocol was registered at PROSPERO: [CRD42021275065].

### Eligibility criteria

#### Inclusion criteria.

1. HCWs, and health science students from Africa continent, were included. HCW was operationally defined as; Doctors, Nurses, Pharmacists, allied health professionals, paramedics, and Healthcare students [i.e., medical students, nurse students etc.].

2. Adults HCWs aged ≥18 years were included.

3. All primary studies such as longitudinal, cohort, case-control and cross-sectional studies reporting COVID-19 vaccine acceptance and hesitancy among HCWs in Africa were included in the current review.

4. Original observational studies published in English were included.

#### Exclusion criteria.

1. General population, other university students, and children were excluded.

2. Non-COVID-19 vaccine acceptance studies.

3. COVID-19 studies reporting animal studies, reviews, commentaries, letter to editors were excluded.

4. COVID-19 vaccine acceptance and hesitancy articles published in other language other than English were excluded

5. COVID-19 acceptability studies among HCWs outside the Africa continent [i.e., Asia, Europe, America, and Australia continents] were also excluded.

## Outcome of interest

The outcome of interest was COVID-19 vaccine acceptance/and hesitancy rate among HCWs in Africa. In addition, the reasons for COVID-19 hesitancy were explored.

## Information sources and search strategies

Medline/PubMed, Google Scholar, Africa Journal Online, and MedRxiv (preprint) were searched. The search was restricted to studies published between January,2020 to September, 2021. The search was limited to articles published in English. Reference lists of articles that met the inclusion and exclusion criteria were reviewed manually to identify additional studies.

Medical Sub-Heading (MeSH) terms and free text were used in the search approach. These terms were combined with the Boolean operators 'OR' and 'AND'. The key terms included; COVID-19, Vaccine, Hesitancy, acceptance, Health care worker, Africa, Sub-Saharan Africa. The full search string is shown in S2 Table.

## Data screening and selection

The data screening and selection involved the following; Two co-authors independently screened the titles and abstracts against the eligibility criteria. Full texts of the articles were then obtained. A disagreement was then resolved by consensus. To ensure that independent reviewers apply the selection criteria reliably, a screening guide was used [18].

## Data extraction and management

Two co-authors extracted the data from the eligible published articles using a pre-tested and standardized excel spreadsheet. Data such as the author's name, year of publication, country, survey period, study design, sample size, HCWs population, acceptance rate, and, reasons for COVID-19 acceptance/hesitancy rate were extracted. Mendeley was used to managed and remove duplicated articles.

## Quality assessment and risk of bias

The Newcastle-Ottawa Quality Assessment tool adapted for cross-sectional studies [19] was used to assess the quality of the retrieved studies. It is graded on 10-point stars. This process was done by two independent reviewers and average was taken as a final score for that particular study. The Newcastle-Ottawa Quality Assessment tool contains three domains. Domain 1 evaluates the methodological quality of each study [5 stars], domain 2 assesses the comparability of the study [2 stars] and domain 3 evaluates the outcome measure and related statistical analysis [3 stars] [19]. Furthermore, the review rated the overall quality of the studies into three; [low risk of bias (7–10), moderate risk of bias (5–6), and high risk of bias (0–4)] [20].

## Data synthesis

Extracted data was exported into Stata (version 16; Stata Cooperation, TX, USA) from Microsoft excel 2013 for all analyses. Due to the presence of heterogeneity [$I^2$ = 96%, p≤0.001], a

meta-analysis using the random effect model was used to pool the COVID-19 vaccine acceptance rate among the HCWs at 95% confidence interval and presented in a forest plot. The presence of heterogeneity among studies was quantified by estimating the variance using the $I^2$ statistics [21]. The $I^2$ takes values between 0 and 100%, and a value of 0% indicates absence of heterogeneity. $I^2$ was interpreted based on Higgins and Thompson classification, percentages of 25%, 50% and 75% was considered as low, moderate and high heterogeneity, respectively [21]. A sub-group analysis was performed based on sub-region (West Africa vs. East Africa vs. Southern Africa vs. North Africa) and type of participants (Healthcare workers vs. Healthcare students). Leave one out sensitivity analysis was performed to examine the effects of a single study on the overall pooled estimate. Publication bias was checked by the funnel plot and Egger's test. The drivers/factors for COVID-19 vaccine non-acceptance/hesitancy among HCWs in Africa were systematically reviewed. A factor/driver for COVID-19 vaccine non-acceptance/hesitancy was eligible if it had been assessed and data from at least two studies were available.

## Results

The electronic search yielded 513 articles; 400 articles remained after the duplicate articles were deleted. After screening the abstracts and titles, 200 articles were removed [i.e., irrelevant to the study]. One hundred and seventy (170) were removed because they were unrelated to the current research. Thirty (30) full-text papers were evaluated for eligibility. Nine papers were removed from the final data synthesis, leaving only 21 articles. The results are displayed in Fig 1.

### Characteristics of the studies

Out of the 21 studies included, 7 were conducted in North Africa, 6 in West Africa, 6 in East Africa and an article each from central and southern Africa. The sample size ranged from 182 to 2133, totaling 14132 participants. The participants were mainly doctors, nurses, medical laboratory scientists, pharmacists, and allied health staff. The first survey was performed in March-April 2020, and the most recent was conducted in March-June 2021. The studies were all cross sectional and published between 2020 and 2021. Majority of the included studies had low-moderate risk of bias [20/21]. The findings are summarized in Table 1.

**Pooled COVID-19 vaccine acceptance rate among HCWs in Africa.** The COVID-19 vaccination acceptance rate was calculated using data from twenty-one (21) studies in Africa. Based on the DerSimonian and Laird random-effects model, meta-analysis revealed a pooled COVID-19 acceptance rate of 46% [95% CI: 37%-54%] (Fig 2). However, there was significant variability among the studies [$I^2$ = 96%, p≤0.001].

### Publication bias assessment

No evidence of publication bias was found after symmetrical inspection using the funnel plot (Fig 3) and Egger's regression test (0.1654).

### Sub-group and sensitivity analysis for COVID-19 vaccine acceptance rate

**Sub-group analysis.** As shown in Table 2, sub-group analysis was based on sub-regions (i.e., North Africa vs. Central Africa, East Africa vs. West Africa). The pooled estimated COVID-19 acceptance rate was 37% [95% CI: 27%-47%] in North Africa, 28% [95% CI: 20%-36%] in Central Africa, 48% [CI: 38%-58%] in West Africa, 49% [95% CI: 30%-69%] in East Africa, and 90% [CI: 85%-96%] in Southern Africa.

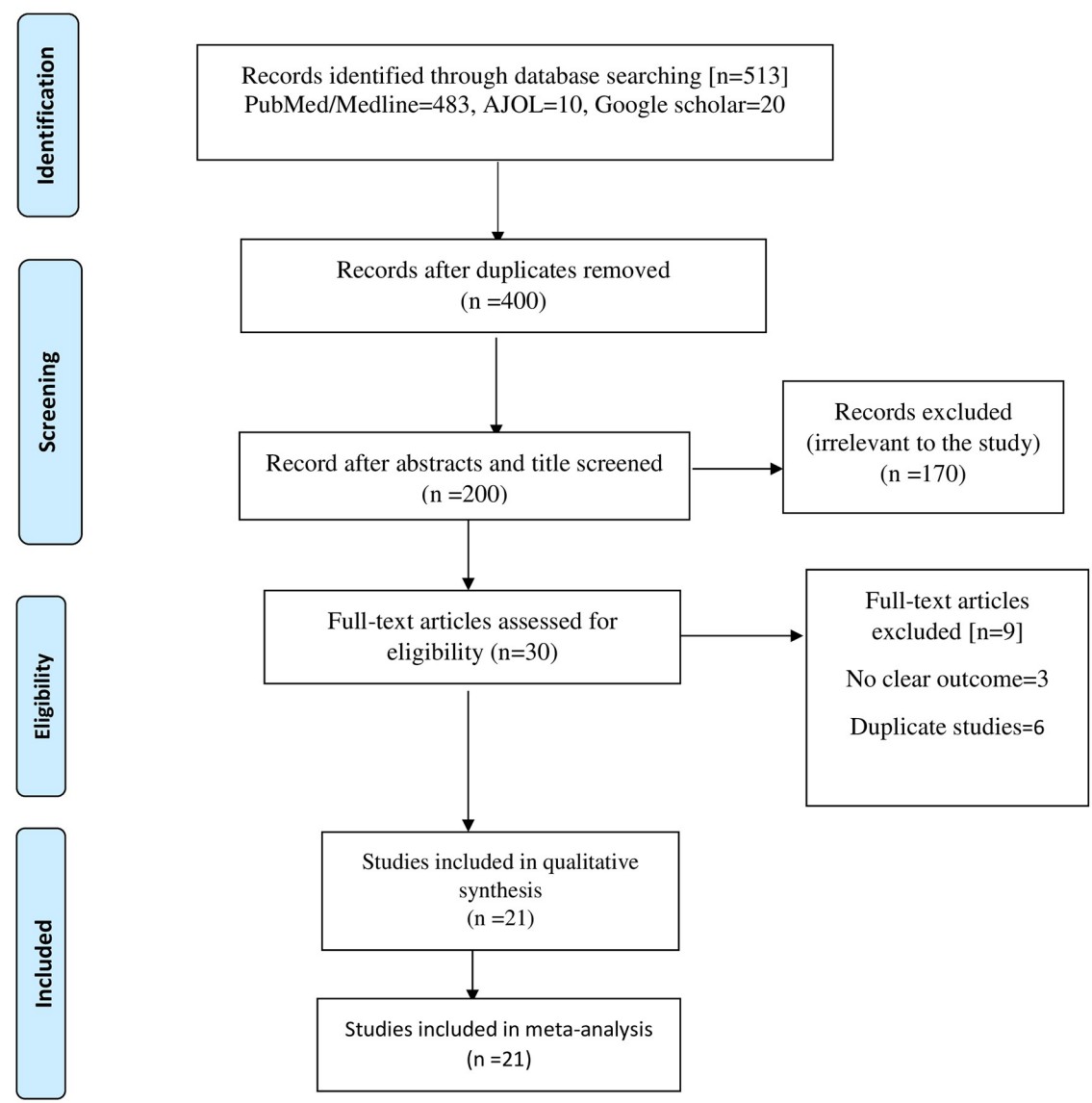

**Fig 1. Preferred Reporting Items for Systematic Review and Meta-Analysis-Adapted flow showing the results of the search.**

Similarly, further stratification by type of participants (health workers vs. health science students). The estimated pooled COVID-19 vaccine acceptance was 48% [95% CI: 38%-57%] for HCWs, and 34% [95% CI: 29%-39%] for the healthcare students [Table 3].

**Sensitivity analysis for COVID-19 acceptance rate.** Leave one out sensitivity analysis was performed to re-estimate the pooled effect on outcome of the remaining studies on the pooled COVID-19 acceptance rate. The results revealed that, no single study had a signifcant impact on the overall result. The pooled acceptability rate of COVID-19 vaccine ranged from 43% [95% CI: 35%-52%] to 47% [95% CI: 38%-55%] (S3 Table).

## Reasons for COVID-19 vaccine non-acceptance among HCWs in Africa

The current systematic review and meta-analysis identified 8 main reasons for COVID-19 vaccine hesitancy among HCWs in Africa. These includes: the side effects of the vaccine [7, 15,

Table 1. Characteristics of the studies [n = 14132].

| Author and Year | Country | Participants | Survey period | Male (%) | Age/Years | Sample Size | Acceptance rate n (%) | Reasons for Vaccine Hesitancy | Quality assessment |
|---|---|---|---|---|---|---|---|---|---|
| Nzaji, 2020 [22] | DR Congo | Doctors, Nurses, Midwives, and Laboratory Technicians | March/April, 2020 | 50.9 | 40.31±11.67, Majority; 25–40 (63%) | 613 | 27.7 | Not stated | Low |
| Fares, 2021 [23] | Egypt | Doctors, Nurses, Pharmacists, Physiotherapists, and Dentists | Dec, 2020-Jan, 2021 | 18.7 | Majority;17–35 (70.4%) | 385 | 21 | 1. Lack of enough clinical trials, and 2. Fear of vaccine's side effects | Low |
| El-Sokkary, 2021 [24] | Egypt | Doctors, Dentists, Pharmacist, and others | Jan, 2021 | 22.4 | NA | 308 | 26 | 1. Severity of COVID-19 2. Vaccine safety | Moderate |
| Agyekum, 2021 [15] | Ghana | Doctors, Nurses/midwives, and Allied health | Jan/Feb, 2021 | 36.8 | Majority; 30–39 (56.0%) | 234 | 39.3 | 1. vaccine safety 2. vaccine side effects 3. Acquiring COVID-19 through vaccination | Moderate |
| Dula, 2021 [7] | Mozambique | | March, 2021 | NA | NA | 566 | 86.6 | 1. Vaccine side effects 2. Made to cause harm 3. Vaccine not effective | Moderate |
| Adeniyi, 2021 [25] | South Africa | Doctors, Nurses, Pharmacists, Allied Health, Support staff | Nov/Dec, 2020 | 18.5 | Majority; 26–55 (79.2%) | 1308 | 90.1 | Not stated | Low |
| Shehata, 2021 [26] | Egypt | Doctors | March/June, 2021 | 40.6 | Majority; 31–40 (71.5%) | 1268 | 24.3 | 1. Vaccine side effects 2. Short duration of Clinical Trial 3. Concerns about safety and efficacy | High |
| Saied,2021 [27] | Egypt | Healthcare Students | Jan, 2021 | 34.8 | 20.2±1.8 | 2133 | 34.9 | 1. Insufficient information about vaccine side effect 2. Insufficient information about the vaccine 3. Insufficient trust from vaccine source | Moderate |
| Kanyike, 2021 [28] | Uganda | Healthcare Students | March, 2021 | 62.8 | Majority; <25 (61.2%) | 600 | 37.3 | 1. Vaccine side effects 2. Misinformation 3. ineffectiveness | Low |
| Ngasa, 2021 [29] | Cameroon | Doctor, Nurse, laboratory technician, Pharmacist, Public health, student, Other | Not stated | 51.8 | 29.1±6.6 | 371 | 45.4 | 1. Efficacy of the vaccine 2. Short duration of clinical trials 3. Adverse effects | Low |
| Aliae, 2021 [30] | Egypt | Doctor, Nurse, laboratory technician, Pharmacist, student, Other | Dec, 2020-Jan, 2021 | 34.9 | Majority; 18–45 (55.0%) | 496 | 45.9 | Not stated | Moderate |
| Alle,2021 [31] | Ethiopia | Doctor, Anesthetists, Nurses, Midwives, Pharmacists, Laboratory Professional | Not stated | 63.6 | Majority; 18–25 (55.0%) | 327 | 42.3 | Not stated | Low |

(Continued)

**Table 1.** (Continued)

| Author and Year | Country | Participants | Survey period | Male (%) | Age/Years | Sample Size | Acceptance rate n (%) | Reasons for Vaccine Hesitancy | Quality assessment |
|---|---|---|---|---|---|---|---|---|---|
| Guangul, 2021 [32] | Ethiopia | Physicians, health officers, nurses, Lab Technicians, Pharmacist, others | Not stated | 69.3 | Majority; 18–29 (58.2%) | 668 | 72.2 | 1. Concerns about safety, 2. Ineffective 3. Acquiring COVID-19 through vaccination 4. Side effects, 5. Short duration of clinical trial | Low |
| Ahmed, 2021 [33] | Ethiopia | All Health professionals | Jan-March, 2021 | 70.2 | Majority;30–39 (54.0%) | 409 | 33.2 | Not stated | Low |
| Annan, 2021 [34] | Ghana | Doctors | Not stated | 49.2 | Majority; 25–30 (83.0%) | 305 | 66.9 | Not stated | Low |
| Adejumo,2021 [35] | Nigeria | Doctors, Nurse, Lab scientist, pharmacist, Physiotherapist, others | Oct, 2020 | 64.3 | 40.0±6.0, Majority 18–40 (72.9%) | 1470 | 55.5 | Not stated | Low |
| Robinson,2021 [36] | Nigeria | All Health professionals | Dec, 2020-Jan, 2021 | 56.7 | Majority; 30–49 (66.6%) | 1094 | 48.8 | 1. Safety, 2. Ineffectiveness 3. Side effects 4. Fear of the unknown | Moderate |
| Oriji, 2021 [37] | Nigeria | Nurses, Lab scientist, Pharmacist, others | April, 2021 | 25.3 | Majority; <36 (47.8) | 182 | 27.4 | 1. To see what will happen (fear) 2. Short duration of Clinical trial 3. Side effects 4. Safety issues 5. Lack of trust in government/ manufacturer | Moderate |
| Khairy, 2021 [38] | Sudan | Doctors, Nurse, Lab scientist, pharmacist, public health, others | March- April 2021, | 46.7 | 35.3±10.6 | 576 | 57 | Not stated | Low |
| Zammit, 2021 [39] | Tunisia | Doctors, Nurses pharmacy, paramedic | Jan, 2021 | 26.6 | 37.4 ±9.5, Majority; <41 (66.1%) | 493 | 48.1 | Not stated | Low |
| Mudenda,2021 [40] | Zambia | Pharmacy student | April, 2021 | 50.3 | Majority; 18–29 (81.2%) | 326 | 24.5 | 1. Side effects 2. Ineffectiveness 3. Safety issues 4. Short clinical trials | Moderate |

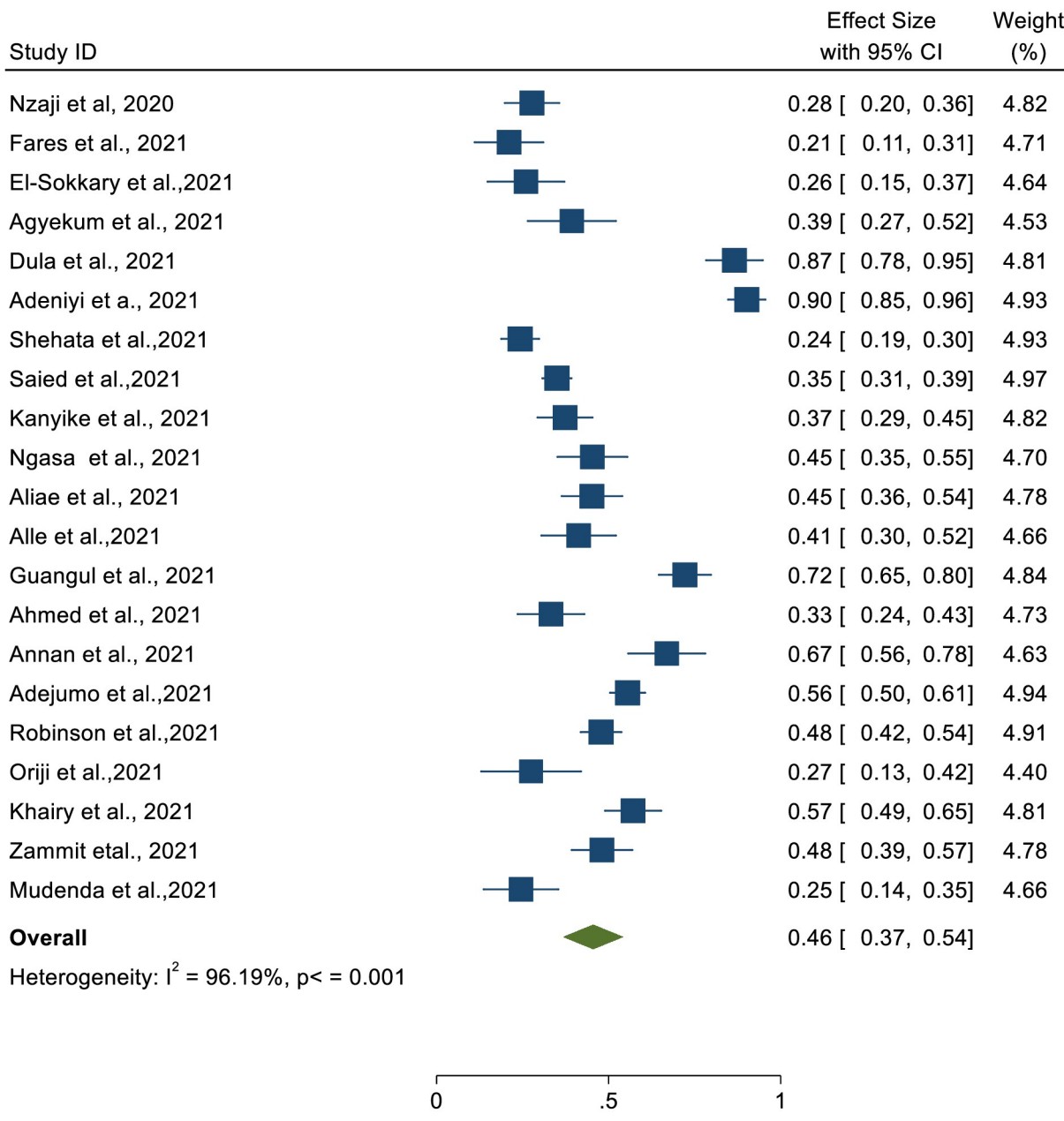

**Fig 2. COVID -19 vaccine acceptance rate among HCWs in Africa.**

23, 26, 28, 29, 32, 36, 37, 40] the vaccine's safety [15, 23, 24, 26, 36, 37, 40], efficacy and effectiveness [7, 26, 28, 29, 32, 36, 40], short duration of the clinical trials [23, 26, 29, 32, 37, 40], COVID-19 infections [15, 32], limited information [27, 28], and lack of social trust [insufficient trust in the vaccine's source, lack of trust from the manufacturers, lack of trust from governments] [27, 37]. The results are summarized in Table 3.

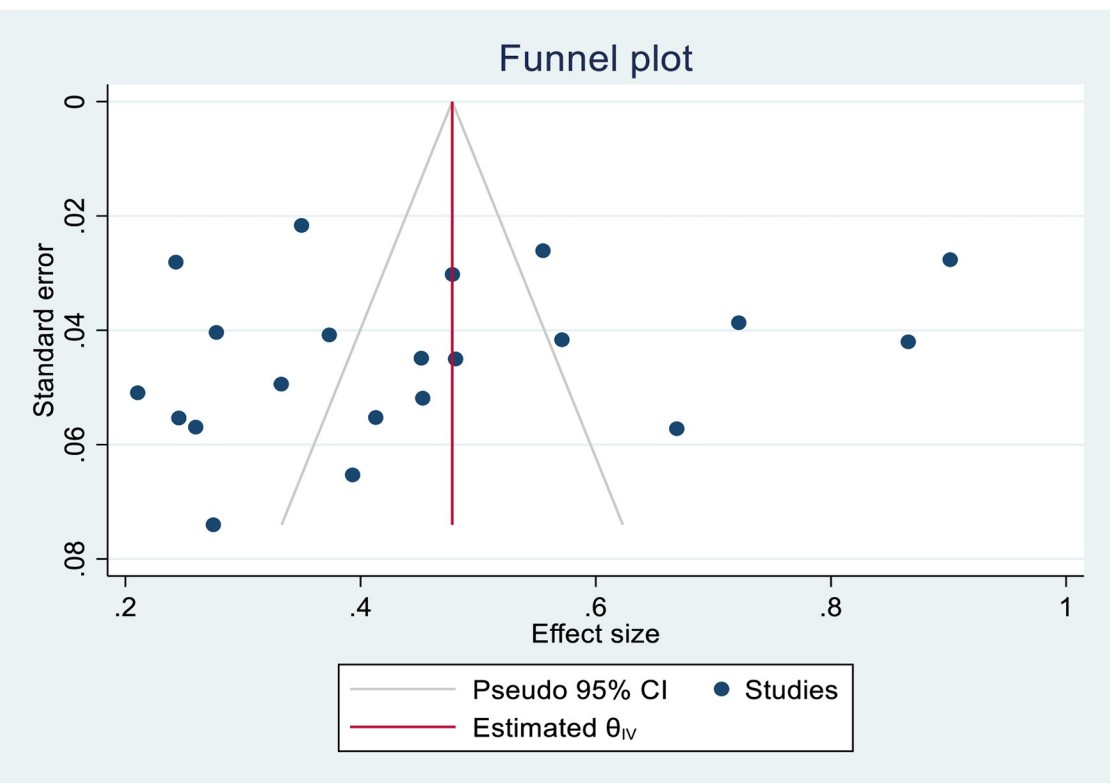

**Fig 3. Assessment of publication bias.**

## Discussion

The systematic review, and meta-analysis was carried out to ascertain the COVID-19 vaccine acceptance rate, and possible reasons for the vaccine's hesitancy amongst HCWs in Africa. The data revealed generally low acceptance of the vaccine amongst HCWs across Africa, and considerable COVID-19 vaccine reluctance. The possible reasons for the vaccine's hesitancy were: the side effects of the vaccine, concerns about the vaccine's safety, efficacy and effectiveness, short duration of the clinical trials, COVID-19 infections, limited information, and social trust.

**Table 2. Pooled COVID-19 vaccine acceptance rate stratified by sub-region and type of participants.**

| Group | Number of datasets | Pooled estimate at 95%CI | $I^2$ [p-value] |
|---|---|---|---|
| **Sub-region** | | | |
| Central Africa | 1 | 28% [95% CI: 20%-36%] | - |
| North Africa | 7 | 37% [CI: 27%-47%] | 92.47% [p≤0.001] |
| West Africa | 6 | 48% [CI: 38%-58%] | 87.31% [≤0.001] |
| South Africa | 1 | 90% [CI: 85%-96%] | - |
| East Africa | 6 | 49% [95% CI: 30%-69%] | 96.50% [≤0.001] |
| **Type of participants** | | | |
| Health workers | 18 | 48% [95% CI: 38%-57%] | 96.14% [≤0.001] |
| Health science student | 3 | 34% [95% CI: 29%-39%] | 37.13% [≤0.001] |

CI = Confidence Interval

**Table 3. Reasons for COVID-19 vaccine hesitancy among HCWs in Africa.**

| Reason | Number of studies | References |
|---|---|---|
| Side effects of the vaccine | 10 | [7, 15, 23, 26, 28, 29, 32, 36, 37, 40] |
| Vaccine's safety | 7 | [15, 23, 24, 26, 36, 37, 40] |
| Efficacy and effectiveness | 7 | [7, 26, 28, 29, 32, 36, 40] |
| Short duration of the clinical trials | 6 | [23, 26, 29, 32, 37, 40] |
| COVID-19 infections | 2 | [15, 32] |
| Limited information | 2 | [27, 28] |
| Lack of Social trust | 2 | [27, 37] |

The overall acceptance rate for the COVID-19 vaccination was 46% [95% CI: 37%-54%]. This is comparable to a previous systematic review and meta-analysis from the western world 51% [14] and higher than a US observational based study of 36% [41]. However, our estimate is lower than prior observational studies conducted in China 86.2% [42], France 76.9% [43], Saudi Arabia 64.9% [44], Canada 80.9% [45], Germany 91.7% [46] and United Kingdom 59% [47]. Low confidence in the vaccine, invasion of media misinformation, conspiracy theories, infodemic, religious beliefs, and possibly past vaccine hesitancy in the continent could all be factors contributing to the low COVID-19 vaccination acceptance rate [48, 49].

The study also revealed an estimated COVID-19 vaccine acceptance rate of 34% [95% CI:29%-39%] by the healthcare students. The estimated value is lower than previous studies in Italy 91.9% [50], US 53% [51], and France 58% [52]. Complacency, exacerbated by low illness risk, and low mortality in the continent since the epidemic began, could be contributing factors in this group of participants [53].

Certain common impediments to the acceptance of the COVID-19 vaccine seemed to be shared by HCWs across the continent. These included, side effects of vaccines, vaccine safety, efficacy and effectiveness of the vaccines, short duration of the clinical trials, the possibility of contracting COVID-19 infection from vaccines, limited information on the vaccines, and lack of social trust (i.e., insufficient trust in the vaccine's source, lack of trust from the manufacturers, lack of trust from governments). vaccine hesitancy is mostly induced by the dissemination of misleading information, primarily through social media platforms and with the assistance of anti-vaccination organizations [54]. Biswas and colleagues conducted a scoping review analysis of 35 studies and found that HCWs worldwide have a 22.5% COVID-19 acceptance rate on average. Their reasons for refusing the vaccination were identical to those revealed in this study [11].

In general, persuading individuals who are vaccine skeptics to change their beliefs is difficult, especially in a continent where there has been a history of vaccination resistance. Nevertheless, it's preferable to concentrate on disseminating positive and factual vaccine information while also strengthening healthcare workers' resistance to fraudulent information. Easy access and mandatory COVID-19 vaccination policies in Africa is a good way to promote COVID-19 immunization uptake. Encouragement of vaccine production within Africa, and comparison of these vaccines with others produced outside the continent, could build more confidence in the safety and efficacy of vaccines among health care workers in the continent. This would involve isolating local strains of the virus to be used in the production of vaccines and the conduction of clinical trials among locals. The outcome of this will be, more tailored interventions to the fight against the pandemic on the continent, and will bring the fight against COVID-19 nearer home. This will also help debunk unfavorable theories about the intents behind the production of vaccines. Finally, the onus is on governments and significant

international stakeholders in the pandemic fight to utilize social media to educate the public, especially HCWs, on facts concerning vaccines in order to help debunk some claims made by conspiracy theorists.

This systematic review and meta-analysis have a number of limitations that should be acknowledged. First, the current review considered only English published papers and as a result some relevant articles maybe missed. Secondary, there was significant heterogeneity across the studies. Nevertheless, to the best of the authors' knowledge, this is the first systematic review and meta-analysis on COVID-19 acceptance and hesitancy rate in Africa. The review used well-validated systematic review and meta-analysis models that are fully compliant with current international standards and recommendations. Sensitivity analyses were performed to determine the robustness of the estimates obtained from the meta-analysis.

## Conclusion

The result of this review revealed generally low acceptance of the COVID-19 vaccine amongst HCWs across Africa. Major drivers and reasons were the side effects of the vaccine, vaccine's safety, efficacy and effectiveness, short duration of the clinical trials, COVID-19 infections, limited information, and social trust. The willingness of HCWs to be vaccinated against COVID-19 acts as a valuable role model for the general public and hence, the misconceptions and barriers to COVID-19 vaccine acceptance amongst these cadre of professionals must be addressed as soon as possible in the continent.

## Supporting information

**S1 Table. PRISMA checklist.**
(DOC)

**S2 Table. Search strategy for the databases.**
(DOCX)

**S3 Table. Leave one out sensitivity analysis.**
(DOCX)

## Author Contributions

**Conceptualization:** Martin Ackah.

**Data curation:** Martin Ackah, Louise Ameyaw, Delali Pearl Afi Asubonteng, Eugene Narkotey Annor, Eunice Abena Kwartemaa Ankapong, Hosea Boakye.

**Formal analysis:** Martin Ackah, Louise Ameyaw, Mohammed Gazali Salifu.

**Investigation:** Martin Ackah, Louise Ameyaw.

**Methodology:** Martin Ackah.

**Resources:** Martin Ackah.

**Software:** Martin Ackah.

**Supervision:** Martin Ackah, Louise Ameyaw, Mohammed Gazali Salifu, Delali Pearl Afi Asubonteng, Cynthia Osei Yeboah.

**Validation:** Martin Ackah, Louise Ameyaw, Mohammed Gazali Salifu, Delali Pearl Afi Asubonteng, Cynthia Osei Yeboah, Eugene Narkotey Annor, Eunice Abena Kwartemaa Ankapong, Hosea Boakye.

**Visualization:** Martin Ackah, Louise Ameyaw, Mohammed Gazali Salifu, Delali Pearl Afi Asubonteng, Cynthia Osei Yeboah, Eugene Narkotey Annor, Eunice Abena Kwartemaa Ankapong, Hosea Boakye.

**Writing – original draft:** Martin Ackah, Louise Ameyaw, Mohammed Gazali Salifu.

**Writing – review & editing:** Martin Ackah, Louise Ameyaw, Mohammed Gazali Salifu, Delali Pearl Afi Asubonteng, Cynthia Osei Yeboah, Eugene Narkotey Annor, Eunice Abena Kwartemaa Ankapong, Hosea Boakye.

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
