## [Decision Letter · Decision Letter 0]

22 Dec 2021

PONE-D-21-31691COVID-19 Vaccine Acceptance among Health care workers in Africa: A Systematic Review and Meta-analysisPLOS ONE

Dear,

Thank you for submitting your manuscript to PLOS ONE. After careful consideration, we feel that it has merit but does not fully meet PLOS ONE’s publication criteria as it currently stands. Therefore, we invite you to submit a revised version of the manuscript that addresses the points raised during the review process. Please submit your revised manuscript by 20th January 2022. If you will need more time than this to complete your revisions, please reply to this message or contact the journal office at plosone@plos.org. Please include the following items when submitting your revised manuscript:A rebuttal letter that responds to each point raised by the academic editor and reviewer(s). You should upload this letter as a separate file labeled 'Response to Reviewers'.A marked-up copy of your manuscript that highlights changes made to the original version. You should upload this as a separate file labeled 'Revised Manuscript with Track Changes'.An unmarked version of your revised paper without tracked changes. You should upload this as a separate file labeled 'Manuscript'.

We look forward to receiving your revised manuscript.

Kind regards,

Muhammad Shahzad Aslam, Ph.D.,M.Phil., Pharm-D

Academic Editor

PLOS ONE

Journal Requirements:

Reviewers' comments:

Reviewer's Responses to Questions

**Comments to the Author**

1. Is the manuscript technically sound, and do the data support the conclusions?

Reviewer #1: Yes

Reviewer #2: Yes

2. Has the statistical analysis been performed appropriately and rigorously? 

Reviewer #1: Yes

Reviewer #2: No

3. Have the authors made all data underlying the findings in their manuscript fully available?

Reviewer #1: Yes

Reviewer #2: Yes

4. Is the manuscript presented in an intelligible fashion and written in standard English?

Reviewer #1: No

Reviewer #2: No

5. Review Comments to the Author

Reviewer #1: Thank you for your research and manuscript. The content of the manuscript is important and needed in today's context around the COVID-19 pandemic. I would suggest major changes for the manuscript regarding grammar, spelling and some broader points regarding the purpose of the research and implications implications of results. Please find below my detailed comments:

1. Grammar and spelling:

- I would recommend authors to review the grammar of the article. There are some sentences throughout the manuscript that are grammatically incorrect, the verb form does match the pronoun.

- There is also the use of pronouns that could be summarized differently, for example authors refer to MA and LA in the methods section, they could just refer to MA and LA as “two authors” or “two researchers.”

-COVID-19 should be capitalized. In many instances it is not capitalized, please correct.

2. Page 5, first paragraph:

I suggest authors to review the way they reference other studies. For example, on page 5, first paragraph; authors cite a study by Sevidzem Wirsty and colleagues. It is common to cite as “Sevidzem Wirsty et al.” I would also suggest to summarize information from other studies instead of adding the exact quote. For example in the same paragraph, the phrase “Vaccine hesitancy, attitude and uptake by health care workers….” In the same paragraph the sentence “Vaccine aversion among the general public has a direct association to vaccine hesitancy among HCWs” does not have a reference, please provide reference.

3. Please complete the protocol at PROSPERO. The protocol that is published does not match the one described in the submitted paper:

- Participants are different in the submitted manuscript and the PROSPERO protocol

4. In the section “Information Sources and search strategies” on page 8: Please include the exact dates of the search (e.g. from January 2020 to Month/year).

5. Figures and tables.

-The resolution of the figures is not high. I can see pixels in all figures, it would be desirable for the image to be clearer.

- All figures are named “Figure 1” please fix the numbers of the figures to match the text.

6. Comments in the discussion section:

- Page 16, 3rd paragraph: The authors claim the estimated value of acceptance rate they found is lower than those in Italy, US and France. I would suggest authors to write what the acceptance rates were in those country to have figures to compare to the African countries they included in the study.

- The manuscript could be improved by providing more information about the implications of the study regarding policy. The authors discuss policies such as easier access to vaccination and mandatory vaccination policies. They could provide information about what works for high vaccination acceptance in Africa, for example in Table 2, the acceptance of vaccination rate is 90%, significantly higher than other African countries. Authors could take the example of South Africa to discuss what works there versus other countries in Africa.

Reviewer #2: The paper is poorly written; specific issues are directly annotated to the pdf of the paper in the attached file. Moreover, the majority of the considered (statistical) approaches, such as the DerSimonian and Laird random-effects model (REM), the I^2 statistic, the funnel plot and the Egger test, are not adequately explained for non-expert readers. Instead, the leave-one-out meta-analysis is well-done enough.

6. PLOS authors have the option to publish the peer review history of their article (what does this mean?). If published, this will include your full peer review and any attached files.

Reviewer #1: No

Reviewer #2: No

---

## [Author Response · Author response to Decision Letter 0]

24 Jan 2022

Response to reviewers’ comments

I sincerely express my warmest greetings to you and your prestigious journal for your comments and feedback. I write on behalf of my co-authors to submit our reply to your astute experienced reviewers' insightful comments. The methodology used follows a point-by-point approach to responding to all comments. Please see below for our response.

Reviewer 1

Comment: Grammar and spelling:

 I would recommend authors to review the grammar of the article. There are some sentences throughout the manuscript that are grammatically incorrect, the verb form does match the pronoun. There is also the use of pronouns that could be summarized differently, for example authors refer to MA and LA in the methods section, they could just refer to MA and LA as “two authors” or “two researchers. -COVID-19 should be capitalized. In many instances it is not capitalized, please correct.

Response: As recommended by the reviewer, the entire manuscript has been thoroughly read once again by all authors and a third independent editor to correct all grammatical errors which has improved the manuscript. Also, the authors have replaced the phrase ‘MA and LA’ with ‘’two authors’ ‘as suggested by the reviewer. Additionally, covid-19 is now capitalized 

Comment: Page 5, first paragraph: I suggest authors to review the way they reference other studies. For example, on page 5, first paragraph; authors cite a study by Sevidzem Wirsty and colleagues. It is common to cite as “Sevidzem Wirsty et al.” I would also suggest to summarize information from other studies instead of adding the exact quote. For example, in the same paragraph, the phrase “Vaccine hesitancy, attitude and uptake by health care workers….” In the same paragraph the sentence “Vaccine aversion among the general public has a direct association to vaccine hesitancy among HCWs” does not have a reference, please provide reference.

Response: Authors have taken the reviewer’s comment into consideration and accordingly paraphrased and summarized the sentence. It now reads ‘’ Sevidzem et al identified and evaluated some probable link to vaccination acceptability in Africa. The factors included vaccination deployment plans, religious practices, vaccine hesitation, proliferation of misinformation, HCW attitudes towards the vaccine, social effects, and supportive environment (12)’. Additionally, ‘’Vaccine aversion among the general public has a direct association to vaccine hesitancy among HCWs’’ has now been referenced.

Comment: Please complete the protocol at PROSPERO. The protocol that is published does not match the one described in the submitted paper: Participants are different in the submitted manuscript and the PROSPERO protocol.

Response: Thanks for the comment. The PROSPERO protocol has been amended now.

comment: In the section “Information Sources and search strategies” on page 8: Please include the exact dates of the search (e.g., from January 2020 to Month/year).

Response: Thanks for the comment. The exact dates for the search have now been included i.e., January, 2020 September, 2021

comment: Figures and tables. The resolution of the figures is not high. I can see pixels in all figures, it would be desirable for the image to be clearer. All figures are named “Figure 1” please fix the numbers of the figures to match the text

Response; The authors thank the reviewer for these observations. The figures and table re-done for higher resolutions. All figures named ‘’Figure 1’’ were software and editorial issues. This will be rectified at the editorial level

comment: Comments in the discussion section: Page 16, 3rd paragraph: The authors claim the estimated value of acceptance rate they found is lower than those in Italy, US and France. I would suggest authors to write what the acceptance rates were in those country to have figures to compare to the African countries they included in the study.

Response: Authors have taken the reviewer’s comment into consideration and accordingly added the COVID-19 vaccine acceptance rate to those countries compared. The paragraph now reads’’ This is comparable to a previous systematic review and meta-analysis from the western world 51% (14) and higher than a US observational based study 36% (40). However, our estimate is lower than prior observational studies conducted in China 86.2% (41), France 76.9% (42), Saudi Arabia 64.9% (43), Canada 80.9% (44), Germany 91.7% (45) and United Kingdom 59% (46)’’……. The study also revealed an estimated COVID-19 vaccine acceptance rate of 34% [95% CI:29%-39%] by the healthcare students. The estimated value is lower than previous studies in Italy 91.9% (47), US 53% (48), and France 58% (49)

Comment: The manuscript could be improved by providing more information about the implications of the study regarding policy. The authors discuss policies such as easier access to vaccination and mandatory vaccination policies. They could provide information about what works for high vaccination acceptance in Africa, for example in Table 2, the acceptance of vaccination rate is 90%, significantly higher than other African countries. Authors could take the example of South Africa to discuss what works there versus other countries in Africa.

Response: Thanks for the comment. While there is no clear policy driving the high vaccination rate in south African on Policy implication, the authors have now discussed policy implication that could drive high vaccination acceptance in Africa, and probably other low- and middle-income countries and the global as a whole. it reads ‘’ it's preferable to concentrate on disseminating positive and factual vaccine information while also strengthening healthcare workers' resistance to fraudulent information. Easy access and mandatory COVID-19 vaccination policies in Africa is a good way to promote covid-19 immunization uptake. Encouragement of vaccine production within Africa, and comparison of these vaccines with others produced outside the continent, could build more confidence in the safety and efficacy of vaccines among health care workers in the continent. This would involve isolating local strains of the virus to be used in the production of vaccines and the conduction of clinical trials among locals. The outcome of this will be, more tailored interventions to the fight against the pandemic on the continent and will bring the fight against COVID-19 nearer home. This will also help debunk unfavorable theories about the intents behind the production of vaccines. Finally, the onus is on governments and significant international stakeholders in the pandemic fight to utilize social media to educate the public, especially HCWs, on facts concerning vaccines in order to help debunk some claims made by conspiracy theorist

Reviewer 2: 

Comment: The paper is poorly written; specific issues are directly annotated to the pdf of the paper in the attached file. Moreover, the majority of the considered (statistical) approaches, such as the DerSimonian and Laird random-effects model (REM), the I^2 statistic, the funnel plot and the Egger test, are not adequately explained for non-expert readers. Instead, the leave-one-out meta-analysis is well-done enough.

Response: The authors would like to thank the reviewer for the annotated corrections. The grammatical errors, spacing, referencing, others have been duly corrected as highlighted in the current manuscript. Additionally, the considered statistical approaches have further been explained. It reads ‘’ Due to the presence of heterogeneity [ I2=96%, p≤0.001], a meta-analysis using the random effect model was used to pooled the vaccine acceptance rate among the HCWs at 95% confidence interval and presented in a forest plot. The presence of heterogeneity among studies was quantified by estimating the variance using the I2 statistics (21). The I2 takes values between 0 and 100%, and a value of 0% indicates absence of heterogeneity. I2 was interpreted based on Higgins and Thompson classification, percentages of 25%, 50% and 75% was considered as low, moderate and high heterogeneity, respectively’’

---

## [Decision Letter · Decision Letter 1]

15 Mar 2022

PONE-D-21-31691R1COVID-19 Vaccine Acceptance among Health care workers in Africa: A Systematic Review and Meta-analysisPLOS ONE

Dear Dr. Ackah,

Thank you for submitting your manuscript to PLOS ONE. After careful consideration, we feel that it has merit but does not fully meet PLOS ONE’s publication criteria as it currently stands. Therefore, we invite you to submit a revised version of the manuscript that addresses the points raised during the review process. Please submit your revised manuscript by Apr 29 2022 11:59PM. If you will need more time than this to complete your revisions, please reply to this message or contact the journal office at plosone@plos.org. Please include the following items when submitting your revised manuscript:A rebuttal letter that responds to each point raised by the academic editor and reviewer(s). You should upload this letter as a separate file labeled 'Response to Reviewers'.A marked-up copy of your manuscript that highlights changes made to the original version. You should upload this as a separate file labeled 'Revised Manuscript with Track Changes'.An unmarked version of your revised paper without tracked changes. You should upload this as a separate file labeled 'Manuscript'.

We look forward to receiving your revised manuscript.

Kind regards,

Muhammad Shahzad Aslam, Ph.D.,M.Phil., Pharm-D

Academic Editor

PLOS ONE

Additional Editor Comments:

Please check the language of the manuscript and request proofreading the paper.

Reviewers' comments:

Reviewer's Responses to Questions

**Comments to the Author**

1. If the authors have adequately addressed your comments raised in a previous round of review and you feel that this manuscript is now acceptable for publication, you may indicate that here to bypass the “Comments to the Author” section, enter your conflict of interest statement in the “Confidential to Editor” section, and submit your "Accept" recommendation.

Reviewer #1: All comments have been addressed

Reviewer #2: All comments have been addressed

2. Is the manuscript technically sound, and do the data support the conclusions?

Reviewer #1: Partly

Reviewer #2: Yes

3. Has the statistical analysis been performed appropriately and rigorously? 

Reviewer #1: I Don't Know

Reviewer #2: Yes

4. Have the authors made all data underlying the findings in their manuscript fully available?

Reviewer #1: Yes

Reviewer #2: Yes

5. Is the manuscript presented in an intelligible fashion and written in standard English?

Reviewer #1: No

Reviewer #2: Yes

6. Review Comments to the Author

Reviewer #1: I thank the authors for their efforts in answering to comments. Some comments were partially addressed. For example, authors have not edited the word COVID throughout the manuscript (either in upper or lower case), some sentences still include grammatical mistakes throughout the manuscript. Re-reviewing the manuscript, I also realized that the inclusion and exclusion criteria for the manuscripts included in the review is poorly detailed.

Overall, I consider the research and results very important, but the manuscript lacks clarity in detailing methods and results and in other sections of the manuscript. The manuscript can improve enormously with a thorough review of grammar and reflection on the order of reporting methods and results.

Reviewer #2: (No Response)

7. PLOS authors have the option to publish the peer review history of their article (what does this mean?). If published, this will include your full peer review and any attached files.

Reviewer #1: No

Reviewer #2: No

---

## [Author Response · Author response to Decision Letter 1]

18 Mar 2022

Response to reviewers’ comments

I sincerely express my warmest greetings to you and your prestigious journal for your comments and feedback. I write on behalf of my co-authors to submit our reply to your astute experienced reviewers' insightful comments. The methodology used follows a point-by-point approach to responding to all comments. Please see below for our response.

Reviewer 1

Comments: I thank the authors for their efforts in answering to comments. Some comments were partially addressed. For example, authors have not edited the word COVID throughout the manuscript (either in upper or lower case), some sentences still include grammatical mistakes throughout the manuscript. Re-reviewing the manuscript, I also realized that the inclusion and exclusion criteria for the manuscripts included in the review is poorly detailed.

Overall, I consider the research and results very important, but the manuscript lacks clarity in detailing methods and results and in other sections of the manuscript. The manuscript can improve enormously with a thorough review of grammar and reflection on the order of reporting methods and results

Authors’ response: The authors would like to use this opportunity to thank the reviewer for these vital observations. Authors have taken the reviewer’s comment into consideration and accordingly edited the word COVID in upper case throughout the manuscript. Moreover, the entire manuscript has been thoroughly read once again by all authors and a third independent editor to correct all grammatical errors which addresses the grammatical concerns raised by the reviewer. Also, the inclusion and exclusion criteria sections has been reviewed. Finally, the order of reporting methods and results follow a standard guidelines as reported in other reviews.

---

## [Decision Letter · Decision Letter 2]

22 Apr 2022

PONE-D-21-31691R2COVID-19 Vaccine Acceptance among Health care workers in Africa: A Systematic Review and Meta-analysisPLOS ONE

Dear Dr. Ackah,

Thank you for submitting your manuscript to PLOS ONE. After careful consideration, we feel that it has merit but does not fully meet PLOS ONE’s publication criteria as it currently stands. Therefore, we invite you to submit a revised version of the manuscript that addresses the points raised during the review process.

We look forward to receiving your revised manuscript.

Kind regards,

Muhammad Shahzad Aslam, Ph.D.,M.Phil., Pharm-D

Academic Editor

PLOS ONE

Journal Requirements:

Additional Editor Comments:

Please correct the mistakes highlighted by reviewer and submit again.

Reviewers' comments:

Reviewer's Responses to Questions

**Comments to the Author**

1. If the authors have adequately addressed your comments raised in a previous round of review and you feel that this manuscript is now acceptable for publication, you may indicate that here to bypass the “Comments to the Author” section, enter your conflict of interest statement in the “Confidential to Editor” section, and submit your "Accept" recommendation.

Reviewer #1: (No Response)

Reviewer #2: All comments have been addressed

2. Is the manuscript technically sound, and do the data support the conclusions?

Reviewer #1: Partly

Reviewer #2: Yes

3. Has the statistical analysis been performed appropriately and rigorously? 

Reviewer #1: Yes

Reviewer #2: Yes

4. Have the authors made all data underlying the findings in their manuscript fully available?

Reviewer #1: Yes

Reviewer #2: Yes

5. Is the manuscript presented in an intelligible fashion and written in standard English?

Reviewer #1: Yes

Reviewer #2: Yes

6. Review Comments to the Author

Reviewer #1: There are several point to address in the manuscript including grammatical changes, please see below:

Comments regarding Introduction:

- In the Introduction, page 4, paragraph 3. Please correct the grammatical mistake from “…..with the world facing an economic downturn and the future uncertain,” to “…with the world facing an economic downturn and an uncertain future,….”

Comments regarding Methods:

- Authors list inclusion criteria for studies, the third criterion is “The total number of workers surveyed” but authors do not specify the total number that they considered appropriate for the inclusion criterion.

- In the section “Exclusion criteria”, authors mention they exclude studies focused on the general population twice, in exclusion criterion 1 and in exclusion criterion 2, please remove redundancy.

-In section “Information sources and search strategies”, please correct the sentence “Reference lists of articles that met the inclusion and exclusion were manually checked to identify extra studies…..” to “Reference lists of articles that met the inclusion and exclusion criteria were reviewed manually to identify additional studies……”

-The study has two objectives: to determine the level of COVID-19 vaccine acceptance and to assess the drivers of COVID-19 vaccine non-acceptance/hesitancy. However, authors do not specify how they systematically assessed drivers of COVID-19 vaccine non-acceptance.

Comments regarding the Results section:

-Additionally, authors should clarify what is social trust, how is social trust defined in the studies they reviewed. Reasons for hesitancy are pretty clear (e.g. safety, effectiveness, duration of clinical trials), except for the concept of social trust. Please clarify in results and discussion.

- In the results section, a subtitle needs to be changed from “Characteristics of included study” to “Characteristics of the studies”.

-Table 2 needs a proper legend. The legend does not specify what the pooled estimate is. A table needs to be able to stand by itself, here the readers do not know if the pooled estimate is the percentage of acceptance or the percentage of hesitancy.

Comments regarding the Discussion section:

-In the discussion section, second paragraph, authors mention factors that contribute to vaccine hesitancy (media misinformation, conspiracy theories etc). Authors do not have a reference for these factors. Did they find these factors in the studies they analyzed? Please add reference or clarify.

-In the discussion section, third paragraph, authors also make a conjecture about why healthcare students show low acceptance rates, but authors do not cite where this information comes from. Was this found in the studies they analyzed? Did they do a review of media articles to make these conjectures? Please add appropriate reference.

Reviewer #2: (No Response)

7. PLOS authors have the option to publish the peer review history of their article (what does this mean?). If published, this will include your full peer review and any attached files.

Reviewer #1: No

Reviewer #2: No

---

## [Author Response · Author response to Decision Letter 2]

26 Apr 2022

Response to reviewers’ comments

I sincerely express my warmest greetings to you and your prestigious journal for your comments and feedback. I write on behalf of my co-authors to submit our reply to your astute experienced reviewers' insightful comments. The methodology used follows a point-by-point approach to responding to all comments. Please see below for our response.

Reviewer 1

There are several points to address in the manuscript including grammatical changes, please see below:

Comments regarding Introduction:

Comment: In the Introduction, page 4, paragraph 3. Please correct the grammatical mistake from “…. with the world facing an economic downturn and the future uncertain,” to “…with the world facing an economic downturn and an uncertain future….”

Response: Thanks for the comment and suggestion. The grammatical error is now rectified as suggested.

Comments regarding Methods:

Comment: Authors list inclusion criteria for studies, the third criterion is “The total number of workers surveyed” but authors do not specify the total number that they considered appropriate for the inclusion criterion.

Response: Thanks for the comment. The third criterion is now deleted to prevent further ambiguity. It is very important to note that the removal of this criterion does not affect the included studies in this review in anyway.

Comment: In the section “Exclusion criteria”, authors mention they exclude studies focused on the general population twice, in exclusion criterion 1 and in exclusion criterion 2, please remove redundancy.

Response: Thanks for the comment and observation. Redundancy is checked and removed as suggested.

Comment: In section “Information sources and search strategies”, please correct the sentence “Reference lists of articles that met the inclusion and exclusion were manually checked to identify extra studies….” to “Reference lists of articles that met the inclusion and exclusion criteria were reviewed manually to identify additional studies……”

Response: Thanks for the comment and suggestion. The sentence is now rectified as suggested.

Comment: The study has two objectives: to determine the level of COVID-19 vaccine acceptance and to assess the drivers of COVID-19 vaccine non-acceptance/hesitancy. However, authors do not specify how they systematically assessed drivers of COVID-19 vaccine non-acceptance.

Response: Thanks for the comment and the observation. We have made a statement to that effect under the data synthesis section. It reads ‘’ The drivers/factors for COVID-19 vaccine non-acceptance/hesitancy among HCWs in Africa were systematically reviewed. A factor/driver for COVID-19 vaccine non-acceptance/hesitancy was eligible if it had been assessed and data from at least two studies were available’’. 

Comments regarding the Results section:

Comment: Additionally, authors should clarify what is social trust, how is social trust defined in the studies they reviewed. Reasons for hesitancy are pretty clear (e.g. safety, effectiveness, duration of clinical trials), except for the concept of social trust. Please clarify in results and discussion.

Response: Thanks for the comment. Concept of social trust is now defined and clarified in the results and discussion. It’s defined as ‘’ insufficient trust in the vaccine’s source, lack of trust from the manufacturers, lack of trust from governments’’.

comment: In the results section, a subtitle needs to be changed from “Characteristics of included study” to “Characteristics of the studies”.

Response: Thanks for the comment and suggestion. The subtitle is now rectified as suggested.

Comment: Table 2 needs a proper legend. The legend does not specify what the pooled estimate is. A table needs to be able to stand by itself, here the readers do not know if the pooled estimate is the percentage of acceptance or the percentage of hesitancy.

Response: Thanks for the comment. The Table 2 legend is rectified elaboratively.

Comments regarding the Discussion section:

Comment: In the discussion section, second paragraph, authors mention factors that contribute to vaccine hesitancy (media misinformation, conspiracy theories etc). Authors do not have a reference for these factors. Did they find these factors in the studies they analyzed? Please add reference or clarify.

Response: Thanks for the comment. The authors have referenced the statement appropriately.

Comment: In the discussion section, third paragraph, authors also make a conjecture about why healthcare students show low acceptance rates, but authors do not cite where this information comes from. Was this found in the studies they analyzed? Did they do a review of media articles to make these conjectures? Please add appropriate reference

Response: Thanks for the comment. The authors have referenced the statement appropriately.

---

## [Editor Report · Decision Letter 3]

6 May 2022

COVID-19 Vaccine Acceptance among Health care workers in Africa: A Systematic Review and Meta-analysis

PONE-D-21-31691R3

Dear,

We’re pleased to inform you that your manuscript has been judged scientifically suitable for publication and will be formally accepted for publication once it meets all outstanding technical requirements.

Kind regards,

Muhammad Shahzad Aslam, Ph.D.,M.Phil., Pharm-D

Academic Editor

PLOS ONE
---

## [Editor Report · Acceptance letter]

10 May 2022

PONE-D-21-31691R3 

COVID-19 vaccine acceptance among health care workers in Africa: A systematic review and meta-analysis 

Dear Dr. Ackah:

I'm pleased to inform you that your manuscript has been deemed suitable for publication in PLOS ONE. Congratulations! Your manuscript is now with our production department. 

Kind regards, 

on behalf of

Dr. Muhammad Shahzad Aslam 

Academic Editor

PLOS ONE